# Can the Output of a Learned Classification Model Monitor a Person’s Functional Recovery Status Post-Total Knee Arthroplasty?

**DOI:** 10.3390/s22103698

**Published:** 2022-05-12

**Authors:** Jill Emmerzaal, Arne De Brabandere, Rob van der Straaten, Johan Bellemans, Liesbet De Baets, Jesse Davis, Ilse Jonkers, Annick Timmermans, Benedicte Vanwanseele

**Affiliations:** 1Human Movement Biomechanics Research Group, Department of Movement Sciences, KU Leuven, 3001 Leuven, Belgium; ilse.jonkers@kuleuven.be; 2REVAL Rehabilitation Research, Hasselt University, 3590 Diepenbeek, Belgium; rob.vanderstraaten@uhasselt.be (R.v.d.S.); annick.timmermans@uhasselt.be (A.T.); 3Declarative Languages and Artificial Intelligence Group, Department of Computer Science, KU Leuven, 3001 Leuven, Belgium; arne.debrabandere@kuleuven.be (A.D.B.); jesse.davis@kuleuven.be (J.D.); 4Department of Orthopaedics, Ziekenhuis Oost-Limburg, 3600 Genk, Belgium; johan.bellemans@zol.be; 5Pain in Motion Research Group (PAIN), Department of Physiotherapy, Human Physiology and Anatomy, Vrije Universiteit Brussel, 1050 Brussel, Belgium; liesbet.de.baets@vub.be

**Keywords:** osteoarthritis, total knee arthroplasty, machine learning, classification, biomechanics, inertial measurement units

## Abstract

Osteoarthritis is a common musculoskeletal disorder. Classification models can discriminate an osteoarthritic gait pattern from that of control subjects. However, whether the output of learned models (probability of belonging to a class) is usable for monitoring a person’s functional recovery status post-total knee arthroplasty (TKA) is largely unexplored. The research question is two-fold: (I) Can a learned classification model’s output be used to monitor a person’s recovery status post-TKA? (II) Is the output related to patient-reported functioning? We constructed a logistic regression model based on (1) pre-operative IMU-data of level walking, ascending, and descending stairs and (2) 6-week post-operative data of walking, ascending-, and descending stairs. Trained models were deployed on subjects at three, six, and 12 months post-TKA. Patient-reported functioning was assessed by the KOOS-ADL section. We found that the model trained on 6-weeks post-TKA walking data showed a decrease in the probability of belonging to the TKA class over time, with moderate to strong correlations between the model’s output and patient-reported functioning. Thus, the LR-model’s output can be used as a screening tool to follow-up a person’s recovery status post-TKA. Person-specific relationships between the probabilities and patient-reported functioning show that the recovery process varies, favouring individual approaches in rehabilitation.

## 1. Introduction

Knee osteoarthritis (OA) is the most common form of osteoarthritis and affects millions of people worldwide [1,2]. Conventional treatment for OA consists of (a combination of) education, medication, promotion of an active lifestyle, and exercise therapy [2,3,4,5]. If these approaches no longer result in an acceptable reduction in pain or disability, the treatment of choice is a total knee arthroplasty (TKA). To assess the impact of OA and TKA surgery on everyday functioning, patient-reported outcome measures (PROMs) are most often used as part of a clinical follow-up. On the other hand, objective follow-up of biomechanical improvement can be used as an additional factor that facilitates precision medicine (including rehabilitation) [6,7,8].

Altered gait biomechanics early after knee replacement is predictive of altered gait biomechanics at 12 months post-TKA [9]. Thus, movement analysis has been suggested as a method that can indicate the need for physical therapy intervention [9]. However, conventional lab-based methods to capture the biomechanical gait pattern of people are time-consuming, requiring specialised equipment and personnel. Fortunately, the combination of the technological improvements in wearable sensors and the development of advanced statistical models (e.g., machine learning) might provide new avenues to monitor and detect aberrant movement patterns in a clinical, and therefore more ecological, settings.

One of the biggest advances of machine learning is its ability to automate the detection of pathological gait features that are too difficult and too complex for conventional statistics or conventional 3D gait analysis that usually evaluates joint function independently from other joints, whereby patterns are less likely to be recognised [10,11]. Previous studies (Laroche et al., 2014 or Jones et al., 2006) used machine learning in osteoarthritis research to distinguish an osteoarthritic biomechanical gait pattern from the gait pattern of an asymptomatic control subject [11,12]. Recently, we used an L1-regularised logistic regression model to distinguish people with (either pre-operative hip or knee) OA from asymptomatic controls [13]. Separate models were created for five activities of daily living and four rehabilitation exercises. Marker-based joint kinematics during ascending stairs, descending stairs, and level walking could accurately classify people with knee OA and asymptomatic controls [13].

This L1-regularised logistic regression model has the advantage that it outputs a measure of certainty to the classification. Given that a person progresses along a continuum after surgery, a hard prediction cannot capture where on this continuum a person’s current recovery is situated. Therefore, models that display a measure of certainty might be more suitable for follow-up purposes. Nevertheless, whether the output of such a learned classification model—i.e., the probabilistic outcome as a measure of classification certainty—can be used to monitor a person’s recovery status post-TKA is still largely unexplored. The use of a learned classification model to monitor recovery status has the potential to serve as a global metric of recovery based on a large number of biomechanical input parameters. One classification model (Cardiff classifier) previously evaluated the probability of belonging to the TKA class, and has been used to detect improvement after a total knee arthroplasty [12]. The Cardiff classifier showed that at six months post-TKA, 23% of the TKA patients were classified as an asymptomatic control based on objective measures (integrated 3D motion capture with force plate data); however, a striking 74% was classified as an asymptomatic control based on subjective outcomes (PROMs) [14]. This indicates that there is a discrepancy between the classification based on PROMs and the classification based on motion capture data. Moreover, it shows that PROMS provide only part of the relevant information. Motion capture measures obtained in this study were limited for clinical use due to methodological difficulties (e.g., need for specialised lab-based data). Fortunately, as mentioned before, the combination of the technological improvements in wearable sensors and the development of advanced statistical models provides new avenues for individuals in a clinical setting to assess and monitor a person’s functional recovery status.

Therefore, this study aims to train a logistic regression model that can differentiate people pre- and post-TKA from control subjects based on kinematic variables extracted from IMU sensors. Logistic regression outputs a score between 0 and 1, representing the probability that a person belongs to a certain class, i.e., how confident the model is that this subject belongs to a particular class [15]. Therefore, we aim to use the probabilistic output of the logistic regression model to examine whether the output of a learned classification model can monitor a person’s recovery status post-total knee arthroplasty (TKA) (i.e., does the movement pattern of people post-TKA evolve towards that of asymptomatic controls). Moreover, this study aims to explore the interaction between patient-reported outcome and the probability of belonging to the TKA class.

Therefore, we tackle two research questions: (1) Can the output of a learned classification model be used to monitor a person’s recovery status post-TKA? We hypothesise that the probabilistic outcome score of a logistic regression model decreases post-TKA (i.e., probability of belonging to the TKA class decreases over time—indicative of normalisation of biomechanical features). (2) Are the outputs of learned classification models related to patient-reported outcome measures? We hypothesise that if the probability of belonging to the TKA class increases (adverse outcome), this coincides with a more unsatisfactory patient-reported outcome. This probabilistic outcome essentially provides a complete gait analysis summarised in a single outcome score between 0 and 1 (similar to the Gait Profile Score often used in a clinical setting [16]). As such, the probabilistic outcome used in this study could be used as a global metric to monitor recovery based on a large number of biomechanical input data as an addition to the already used PROMs. Moreover, this global metric might also be relevant in determining the end of a rehabilitation trajectory.

## 2. Materials and Methods

### 2.1. Participants

This is an exploratory study based on a secondary analysis of a more extensive prospective follow-up (study was approved by the local ethics committee of the university hospital Leuven, S59857). In this controlled laboratory-based study, 39 people participated: 20 asymptomatic controls and 19 people with unilateral knee OA. The people with knee OA received a total knee arthroplasty, and 17 were re-evaluated at 6 weeks, and at 3, 6, and 12 months post-TKA. One participant dropped out at 6 weeks after having received a double prosthesis, and a second participant dropped out at 12 months post-TKA due to a symptomatic herniated disk in the lower back. People with knee OA were recruited from two local hospitals (Jessa Hospital, Hasselt, Belgium and Ziekenhuis Oost Limburg, Genk, Belgium). The asymptomatic control subjects were recruited from a local senior’s network in Leuven. Inclusion and exclusion criteria for both groups are in Table 1. Before starting the study, participants provided written informed consent.

### 2.2. Study Protocol

Based on Emmerzaal et al., (2022) ascending stairs, descending stairs, and level walking are the three activities that best distinguish people with end-stage knee OA from asymptomatic controls [13]. Therefore, the participants were asked to perform these three activities (i.e., level walking, ascending, and descending stairs), while being equipped with 17 inertial measurement units (IMUs) (MVN BIOMECH Awinda Xsens Technologies, sampling at 60 Hz; software version MVN Awinda 4.4) [17]. These sensors were placed on all body segments except hands (i.e., both feet, lower legs, upper legs, pelvis, sternum, shoulders, upper arms, lower arms, and head) according to the Xsens guidelines (MVN User Manual, 2019). Participants were instructed to walk or navigate the stairs at a self-selected speed; see van der Straaten et al., (2018) for details of the protocol [18]. All activities were performed barefoot. Pre-operatively and for the control subjects, a standardised 4 step staircase was used. For the follow-up measures, a standardised 3 step staircase was used. Step height and tread length was identical in both staircases (height = 0.16 m and length = 0.31 m).

### 2.3. Data Processing

Joint angles were directly extracted from the Xsens graphical user interface (MVN Awinda studio 4.4) [17]. Similar to the workflow of Emmerzaal et al. (2022), only a single middle stride was extracted for each leg and used for further processing [13]. Step detection for walking was performed using the peak acceleration of the anterior-posterior raw acceleration signal [19]. For each detected step, we determined left-right steps using the medio-lateral acceleration signal [19]. To extract the stride from the stairs data, we used the local maxima of the tibia acceleration in the x-direction that followed a local minimum. The peak tibia acceleration was found to be the most robust method to segment stairs data based on the comparison of the step detection based on the peaks in the acceleration signal with the step detection using force plate data. Joint angles from the spine, hip, knee, and ankle of the middle stride were extracted and filtered at 6 Hz to match conventional filtering of joint kinematics used in our previous work [13] and time normalised to 100% stride time.

### 2.4. Feature Construction

The features during all evaluation moments for each subject during the three types of exercise were derived from the kinematic timeseries data using a multivariate feature construction tool, TSFuse (version 1.0dev) [20]. Feature construction was performed in an unsupervised manner (meaning this was conducted without knowledge in which group each participant belonged to). Table 2 shows an overview of kinematic data used as input to TSFuse package. Each input is represented as a view that contains a univariate timeseries for each trial. Based on these time series, TSFuse generates new timeseries using time series fusion operations (e.g., computing the ratio of two time series). To avoid generating redundant timeseries, the system only adds a new timeseries if the correlation to any of the other timeseries is lower than a given threshold. We set this threshold to its default value (0.99). The system then automatically extracts features from the input and generated time series. We used the full setting of TSFuse, which includes all features implemented in the system. These features contain statistics such as the mean, variance, minimum and maximum of a timeseries, as well as more complex features such as Fourier coefficients and linear trend statistics. A complete overview of all implemented features is shown in [20]. After extracting the features for each trial, the features were averaged per subject per activity. Note, for the asymptomatic controls, the features were averaged over both legs. In people with knee OA and after TKA, only the affected/operated side was considered.

### 2.5. Statistical Analysis

Based on the features derived from the kinematic data, we used a leave-one-out cross validation machine learning pipeline that predicted the probability that a given person belongs to the TKA class. Since different activities may require different models, we trained a separate model for each activity. The training phase had two steps:Normalisation: Because of inter-individual differences such as body mass and length, the features of different subjects may have different ranges. To account for such differences, we normalise the features using standard scaling. Specifically, we transform each feature as follows:
(x−mean(x))/std(x)
where *x* is a vector containing the value for each subject and each activity in the training set.Training with hyperparameter tuning: We train a logistic regression model that takes the normalised features as input. Since the number of constructed features is large (25,958 features) in comparison to the number of instances (39 subjects), we select a small set of features by using L1-regularisation, which sets the coefficients of irrelevant and redundant features to zero. We tune the regularisation parameter by training different logistic regression models using five-fold cross-validation on the training set. Specifically, we evaluate ten different values for the regularisation parameter scaled logarithmically between 0.01 and 1. For each value, we evaluate the average area under the ROC curve (AUC) over the five-folds and select the value with the highest average AUC. The final model is then trained on the full training set using the best regularisation parameter. The model was trained using the LIBLINEAR [21] optimiser with a maximum of 100 iterations.

After training, we can evaluate the model on unseen data, i.e., data collected from different subjects or at different points in time. The evaluation phase has the following two steps:Normalisation: Similar to the training phase, we normalise the feature values. We apply the same transformation as for the training data using the mean and standard deviation computed from the training set.Prediction: We apply the trained logistic regression model to the normalised features. The model returns a score between zero and one. Scores close to zero mean that the model is confident that the person is an asymptomatic control, and scores close to one mean that the model is confident that this person is a person and will receive or has received a TKA. Decision boundary is set at 0.5, scores above the 0.5 are classified as TKA and below the 0.5 are classified as asymptomatic.

The analysis is performed in Python (version 3.8.5). We use the scikit-learn library (version 0.21.2) to normalise the features and train the logistic regression models [22].

To analyse whether the considered features within the model capture changes in movement pattern post-TKA we explore two types of input features: (1) model developed on input features measured pre-TKA; to deploy the model on all other time points (i.e., six weeks, three-, six-, and 12 months post-TKA) for each activity; and (2) model developed on input features measured at 6 weeks post-TKA; to deploy the model on the remaining evaluation moments post-TKA (i.e., 3-, 6-, and 12 months post-TKA). For all models, we will first check the model’s ability to differentiate the TKA class from the asymptomatic control class with the classification accuracies of each model (i.e.,the ratio of correct predictions to the total number).

To explore how the participants’ probability relates to patient-reported outcome measures, we used the outcome scores of the activity of daily living (ADL) subsection of the Knee Injury and Osteoarthritis Outcome Score (KOOS) at the different evaluation moments. The ADL subsection was chosen because it evaluates the perceived functioning in daily life—amongst others, during ascending and descending stairs. First, we calculated the change in probability and the change in patient-reported outcome between 6 weeks and 12 months post-op. Thereafter, we calculated the Spearman’s rank correlation on the change scores to get the group correlations. Lastly, we calculated individual Spearman’s rank correlation coefficient between the probability of belonging to the TKA class and the patient-reported functioning per participant over time to get the individual responses. A correlation coefficient between 0–0.25 was considered weak, from 0.25–0.5 fair, 0.5–0.75 moderate, and 0.75–1.0 strong.

## 3. Results

### 3.1. Classification Accuracies

We found that all models had high classification accuracies, ranging from 91.67% to 100%, indicating that all our models are able to differentiate the TKA class from the asymptomatic class (Table 3).

### 3.2. All Activities, Models Trained Pre-Operative Data

Figure 1 shows the heatmaps for the classification model trained on pre-operative data and deployed on all post-TKA re-evaluation moments. For ascending and descending stairs, we observed a decrease in probability to belong to the TKA class in only 27% of our participants. This is visualised in five participants that show a change from darker to lighter colours in the heatmap shown in Figure 1. The majority (72% of our participants), however, does not show a change in the probability score of belonging to the TKA class. However, stair negotiation might be unsuitable for monitoring progress due to the substantial number of participants that could not negotiate stairs at six weeks post-op (white fields in Figure 1). Contrarily, using data from level walking, we observe a large decrease in the probability of belonging to the TKA class between pre- and post-TKA probabilities where all individuals are classified as asymptomatic at six weeks post-op. Due to the substantial number of people unable to negotiate stairs, the remainder of the analysis will only consider level walking, i.e., the model is trained on early post-op data and relates the probability scores with patient-reported functioning.

### 3.3. Level Walking, the Model Trained on Post-Operative Data

Using the model trained on six weeks post-TKA data, we see more minor deviations in the probability of belonging to the TKA class at various stages of recovery (Figure 2). This decrease in the probability of belonging to the TKA class is visualised in a gradual shift from darker to lighter colours in all participants. At 12 months post-TKA, 38% of our participants are now classified as asymptomatic (probability below the 0.5) and have improved their movement pattern to resemble that of asymptomatic controls—indicative that 62% still have an aberrant gait pattern at 12 months post-TKA. Showing that, if we would have used the binary classification as the output, we would have missed the improvements made by the people who do not cross the arbitrary 0.5 threshold.

### 3.4. Probability of Being in the TKA Class Related to Reported Functioning

For the total group, we found a fair correlation between a change in probability of belonging to the TKA class between 6 weeks and 12 months (ρ = −0.37) with a change in ADL score. However, this result was not-significant (*p* = 0.15). Concerning the individual correlations over time, we found a strong (ρ > −0.75) and moderate (ρ = 0.5–0.75) negative correlation between the probability of belonging to the TKA class and the KOOS ADL outcome score in 55.6% and 11.1% of our participants, respectively (Figure 3). In the remaining 33.3%, we only found a fair or weak negative correlation. Nevertheless, from the individual plots in Figure 3 we can also observe that the responses to recovery are person specific—i.e., not everyone shows a clear decrease in probability of belonging to the TKA class. In the people with a strong negative correlation (rows 1 and 2 of Figure 3), we see an apparent coupling between a decrease in the probability of belonging to the TKA class and an increase in the patient-reported functioning score. However, a good outcome on patient-reported functioning (e.g., score > 90) does not necessitate a classification as asymptomatic based on biomechanical gait data (e.g., koa_pp08—row 1 column 3, koa_pp06—row 1 column 4 of Figure 3). Indicating that recovery is a multidimensional process and the two methods might measure a different construct of the recovery process.

## 4. Discussion

This study aimed to explore whether the output of a learned classification model can be used to monitor a person’s recovery status post-TKA and whether these biomechanically-related outputs reflect patient-reported functioning. Our results show that the model’s ability to detect biomechanical changes depends on the activity and the data used to develop the logistic regression classification model: level walking was able to monitor a person’s recovery status when early post-operative IMU-based kinematic data was used to create the model (Figure 2). Models trained on level walking pre-operative data are less applicable for capturing changes during re-evaluation moments post-TKA (Figure 1). Using data from stair negotiation might be unsuitable for monitoring purposes due to the substantial number of participants that could not negotiate stairs at all timepoints. Moreover, the trends in the probability score of the logistic regression model trained on level walking six weeks post-TKA is related to trends in the patient-reported functioning for 67.3% of our participants. However, inter-individual differences in responses during recovery are present.

### 4.1. All Activities, the Model Trained Pre-Operative Data

The logistic regression models trained on pre-operative data are less appropriate for monitoring purposes. The substantial number of subjects (33%) that are unable to negotiate stairs at six weeks post-TKA makes employing this classification model more difficult in clinical practice.

Level walking is an activity that all participants were able to perform. However, the pre-operative model was not able to detect biomechanical changes at follow-up since all individuals were classified as asymptomatic as early as six weeks post-TKA. This result is in contrast with the results of Worsley et al., (2016) who used the Cardiff classifier trained on pre-operative walking data and showed that only 23% retained asymptomatic gait at 6 months post-TKA [14]. Moreover, from a clinical perspective, it is improbable that all our participants function as asymptomatic individuals six weeks after major surgery. Especially considering that previous research found that biomechanical differences in gait pattern compared to asymptomatic controls are still present at 12 months post-TKA [9,23,24], making Worsley’s results more probable [14]. Possibly, the most relevant features in differentiating people with pre-operative end-stage OA from asymptomatic controls are no longer relevant post-TKA—e.g., relevant features might have been resolved with the surgical procedure. Thus, if the goal is to classify the gait pattern of end-stage OA people, features from pre-operative level walking kinematics show perfect accuracy. However to monitor biomechanical changes in the gait pattern post-TKA, this model is less relevant because we are possibly dealing with a different population—i.e., post-TKA instead of end stage OA.

### 4.2. Level Walking, the Model Trained on Post-Operative Data

In contrast, the model trained on early post-operative (6 weeks post-TKA) level walking data was able to monitor biomechanical changes post-TKA at the remaining evaluation moments (i.e., three, six, and 12 months). We can observe more minor deviations in the probability of belonging to the TKA class at the different time points during recovery, which results in a gradual shift from darker to lighter colours (Figure 2). At 12 months post-TKA, 38% of our participants were classified as asymptomatic. This is a slightly higher percentage than Worsley et al. (2016), who found that only 23% of the people with TKA improved using a model trained on pre-operative kinematic, kinetic, and spatiotemporal gait data [14]. However, Worsley et al. (2016) re-evaluated their participants six months post-TKA compared to the one year post-op considered in our study [14]. This longer follow up period might clarify the difference with Worsley’s results.

### 4.3. Interaction between Probability and Patient-Reported Outcome

We explored whether the changes in probabilities of belonging to the TKA class were related to changes in patient-reported functioning between 6 weeks and 12 months post-TKA, and we only found a fair non-significant negative correlation (ρ = −0.37) between the two when considering the group. Indicating that there is only a slight relationship between a larger improvement between 6 weeks and 12 months post-TKA in patient reported functioning, and a larger decrease in the probability of belonging to the patient class. However, when we explored the individual relationships, we found moderate to strong negative correlations between the probability of belonging to the TKA class and patient-reported functioning in 66.6% of our participants. This indicates that a decrease in the probability of belonging to the TKA class was related to an increase in reported functioning for those individuals. However, in 33.3% we only found weak to fair negative correlations. The inter-individual responses to TKA might explain the weaker correlation found when using the whole group. Highlighting the importance of a more individualised approach within a clinical setting.

A more in-depth investigation of the different recovery processes (Figure 3) reveals that despite the strong correlations, an excellent score on the KOOS does not always correspond to a classification of being asymptomatic (biomechanical recovery is lacking?). Despite not reporting any issues in daily life, an aberrant mechanical movement pattern has been related to the onset and progression of OA [25,26,27], illustrating the additional value of monitoring biomechanics alongside patient-reported functioning. A similar phenomenon is present in the opposite direction, where a classification of being asymptomatic does not always correspond to an excellent KOOS score, showing that recovery is a multifactorial process that includes all sides of the biopsychosocial model. These results and those from previous studies (e.g., [6,7,8]) illustrate the additional value of monitoring biomechanics alongside the already-used patient reported outcome measures in a clinical setting to allow for individualised rehabilitation.

Previously conducted validation studies of joint angles provided by the Xsens’ proprietary software have suggested that angles in the frontal and transversal plane are less valid than those calculated in the sagittal plane (see, e.g., [28]). However, we do not believe that the accuracy of the joint angles affected the machine learning model. Because even though the absolute angles might not be valid compared to 3D Motion Capture, we were still able to classify the gait pattern of our participants correctly, and in this case, that is the aim of this study.

Since we only used kinematic data measured through wearable technology (i.e., directly exported from the Xsens MVN 4.4 software [17]), this method is directly translatable to the clinical setting and overcomes the dependence on force plate data needed for the Cardiff classifier [12,23] and conventional 3D biomechanical analysis. Hence, this method can be used as a tool to follow up progress after a TKA and as a screening tool to identify people who, from a biomechanical perspective, are not improving.

### 4.4. Limitations

There are some limitations to this work concerning its clinical use and generalisability. Machine learning allowed us to generate and extract features derived from different sensors to obtain a global metric able to monitor the movement pattern of an individual. However, as we used features extracted from TSFuse, we gain little to no physiological insight into the causes of the progress. Indicating that this metric cannot define specific rehabilitation goals. Therefore, the the probabilistic outcome of the learned classification model can be used as a tool to detect problems in the movement pattern, nevertheless the original Xsens data or a more in-depth physical assessment should be used to detect the causes of poorer rehabilitation outcomes. Moreover, we only included people with unilateral end-stage knee OA without comorbidities. Hence, how well this model functions for people with bilateral prosthesis or comorbidities needs to be investigated further. Because machine learning works well when the test dataset has the same type of input data (i.e., same feature space and distribution) as the training set used to develop the model [29], models should be retrained when the input data changes. Transfer learning methods, where knowledge is transferred from one domain to another, could be beneficial in using previously-trained models and adapting them to different OA groups instead of collecting data from large sample sizes [29].

Alongside the limited heterogeneity, we also have a limited number of participants. This allowed us to investigate individual recovery processes, however, we need further research with more participants to indicate when a change in probability of belonging to the TKA class leads to a clinically relevant change. Moreover, it should be noted that the individual Spearman’s rank correlations were calculated over four data points, i.e., 6 weeks, 3-, 6-, and 12 months post-TKA.

## 5. Conclusions

This study found that the ability of the classification model to monitor a person’s recovery status depends on the activity (i.e., level walking, ascending or descending stairs) and the data used (i.e., pre-op or six weeks post-op data) to develop the logistic regression classification model. Level walking but not stair negotiating was sensitive to changes at 3, 6 and 12 months post-TKA when six weeks post-operative data was used to train the model. Models trained on pre-operative data were unable to distinguish the level of recovery at the post-TKA re-evaluation moments. The individual relationships found between the probabilistic outcome measures and patient-reported functioning showed that the recovery process varies between individuals, favouring individual approaches in rehabilitation. The probabilistic outcome of the learned logistic regression model can potentially be used as a global metric to follow up functional recovery status post-TKA. We believe that this global metric can be used to flag people that are not recovering from a biomechanical point of view.

## Figures and Tables

**Figure 1 sensors-22-03698-f001:**
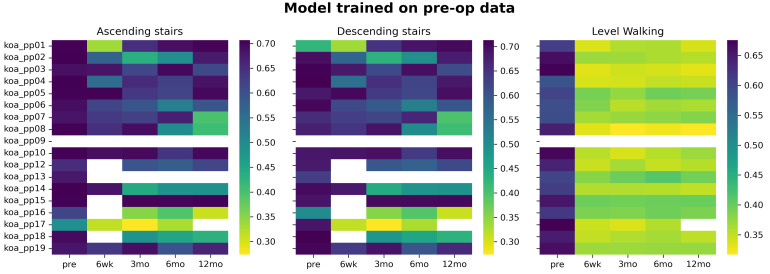
Heatmap of the probability of belonging to the patient class for each individual (subject identifiers are on the y-axis). Darker colours indicate a high probability of belonging to the TKA class, and the lighter the colours lower the probability. The model was developed on pre-operative data of either ascending stairs, descending stairs, or level walking.

**Figure 2 sensors-22-03698-f002:**
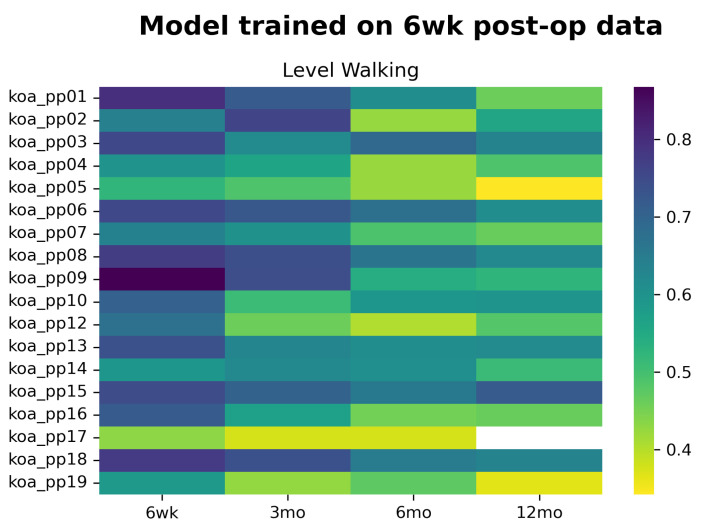
Heatmap of the probability of belonging to the patient class for each individual. Darker colours indicate a high probability of belonging to the TKA class, and the lighter the colours lower the probability. The model was developed on post-operative data of level walking.

**Figure 3 sensors-22-03698-f003:**
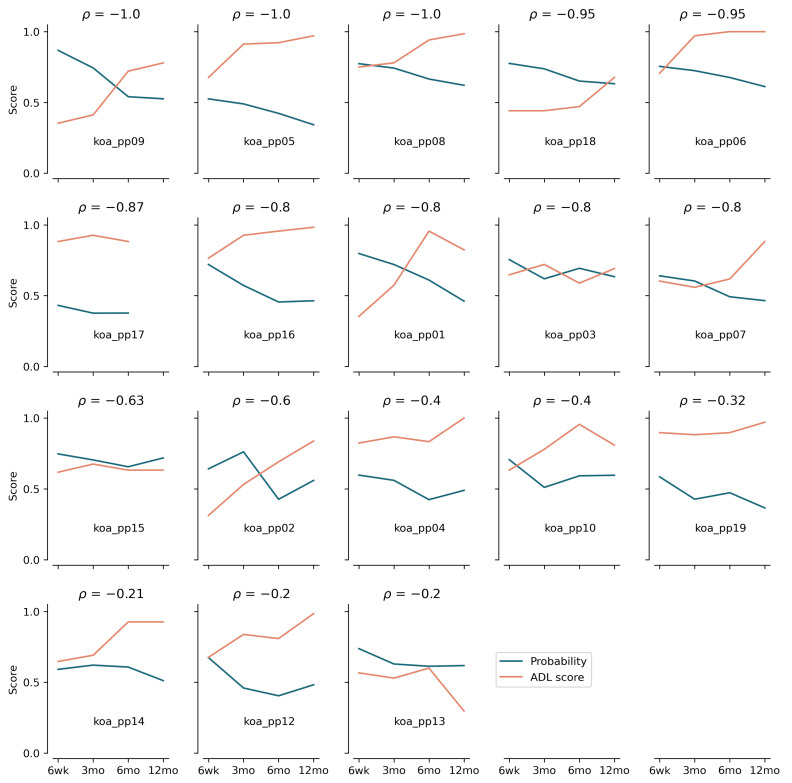
Evaluation time-curves of the probability of belonging to the TKA class (blue) and the patient-reported outcome (orange) per participant. The model used is trained on post-operative walking data and deployed on post-op walking. The headers of each subplot show the correlation coefficient. The subplots are sorted on highest correlation to lowest correlation.

**Table 1 sensors-22-03698-t001:** Inclusion and exclusion criteria for the participants.

Healthy Population	Patient Population
Inclusion	Inclusion
Aged between 50–75 years old Understand the Dutch language Able to walk 10 m and ascent/descent the stairs	Aged between 50–75 years old Understand the Dutch language Diagnosed with hip or knee OA Awaiting of total hip of knee replacement surgery Able to walk 10 m and ascent/descent the stairs
Exclusion	Exclusion
Diagnosed with musculoskeletal or neurological disorders Pain in hips, knees or ankles, which affect normal movement	Corticosteroid injection 3 months before inclusion to the study Diagnosed with symptomatic hip or knee OA on the contralateral knee Joint replacement in other lower limb joints Symptomatic degenerative disorders in other lower limb joints Neurological conditions that could alter movement pattern History of pathological osteoporotic fractures (in hip, knee or ankle joints)

**Table 2 sensors-22-03698-t002:** Timeseries used as input in TSFuse.

Kinematics
Spine lateral bending
Spine flexion/extension
Spine rotation
Hip flexion/extension
Hip abduction/adduction
Hip internal/external rotation
Knee flexion/extension
Knee abduction/adduction
Ankle plantar/dorsiflexion

**Table 3 sensors-22-03698-t003:** Classification accuracies of all models.

Activity	Model Trained on	Classification Accuracy
Ascending stairs	Pre-TKA data	93.33%
Descending stairs	pre-TKA data	96.67%
Level walking	pre-TKA data	100%
Ascending stairs	6 weeks post-TKA data	95.83%
Descending stairs	6 weeks post-TKA data	91.67%
Level walking	6 weeks post-TKA data	94.58%

## Data Availability

The data that support the findings of this study are not openly available due to GDPR guidelines (i.e., human motion data), and are available from the corresponding author upon reasonable request.

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
