# Peer review of "Can the Output of a Learned Classification Model Monitor a Person’s Functional Recovery Status Post-Total Knee Arthroplasty?"

_sensors, 2022, doi:10.3390/s22103698_

Round 1

Reviewer 1 Report

Please find comments attached.

Author Response

Please see attatchment

Reviewer 2 Report

In the submitted manuscript, the authors developed a logistic regression classification model to discriminate between healthy and pathological (i.e., osteoarthritic) gait patterns. 39 participants (20 asymptomatic controls, 19 with unilateral knee osteoarthritis) were outfitted with a full IMU array that provided joint angle data from the IMU manufacturer’s proprietary software. Participants performed three activities: 1) level walking, 2) stair ascent, and 3) stair descent. For the participants with osteoarthritis, these activities were performed before a total knee arthroplasty (TKA) as well as 6 weeks, 3 months, 6 months, and 12 months after the procedure. Features from the joint angle time series were extracted with TSFuse, an unsupervised multivariate feature construction tool. Two models were trained, the first on pre-TKA data and the second on the 6 weeks post-TKA data. The results suggest that both models are capable of detecting change in the TKA patients as they recover, but the models trained on pre-TKA data had inferior performance. The authors also found that the participants’ perceptions of their recovery (i.e., patient-reported outcome measures, PROMs) was not entirely captured by their gait, which suggests recovery is a more involved process than what can be described by their biomechanical ambulation patterns.

There are a few major revisions that should be addressed prior to publication. In Table 3 on page 6, the highest accuracy achieved by their classification models was achieved for level walking trained on pre-TKA data. This result seems to contradict statements made later in the manuscript, i.e., lines 255-256 on page 9, “The logistic regression models trained on pre-operative data are less appropriate for monitoring purposes.” How can the model have perfect accuracy and be less appropriate? Also, in the Introduction, the authors make a point to highlight that recovery is a continuum (i.e., the paragraph spanning lines 48-69). However, the logistic regression approach adopted here converts that continuous spectrum into discrete classes (asymptomatic control, has/will receive a TKA). The authors are encouraged to reframe their introduction to resolve this contradiction. Finally, the authors also make a point of highlighting that predictions about participants’ gait are not strongly correlated with PROMs. Clinically, what does it mean for patients to report good outcomes but fail to be classified as asymptomatic or vice versa?

A few minor revisions are also listed below:

  • How was classification accuracy defined? Meaning, was it the ratio of correct predictions to total predictions, the ratio of true positives/true negatives to total predictions, etc.?
  • What was the size of the training dataset compared to the validation dataset?
  • Please confirm in the caption for Figure 1 on page 6 that the y-axis are subject identifiers.
  • Can the authors confirm for the reader that the features extracted by TSFuse do not necessarily have intuitive physiological meaning?
  • Previously conducted validations of the joint angles provided by Xsens’ proprietary software have suggested that abduction/adduction and internal/external rotation angles are considerably less reliable than flexion/extension angles (see for example, the reference included below). Would this decrease in reliability affect the quality of the features extracted for the logistic regression models to be trained on?
    • Zhang, J.T., Novak, A.C., Brouwer, B. and Li, Q., 2013. Concurrent validation of Xsens MVN measurement of lower limb joint angular kinematics. Physiological measurement34(8), p.N63.

Overall, this manuscript was relatively well-written and communicated. It is the opinion of this reviewer that the manuscript be accepted after major revisions.

Round 2

Reviewer 2 Report

My comments have been addressed. I look forward to seeing the authors' work come to fruition.